# Non-Myopic Multi-Objective Bayesian Optimization

**Syrine Belakaria\***  
*Stanford University*  
*syrineb@stanford.edu*

**Alaleh Ahmadianshalchi\***  
*Washington State University*  
*a.ahmadianshalchi@wsu.edu*

**Barbara E Engelhardt**  
*Stanford University*  
*barbarae@stanford.edu*

**Stefano Ermon**  
*Stanford University*  
*ermon@cs.stanford.edu*

**Janardhan Rao Doppa**  
*Washington State University*  
*jana.doppa@wsu.edu*

**Reviewed on OpenReview:** *https://openreview.net/forum?id=2e1aZZd88C*

## Abstract

We consider the problem of finite-horizon sequential experimental design to solve multi-objective optimization (MOO) of expensive black-box objective functions. This problem arises in many real-world applications, including materials design, where we have a small resource budget to make and evaluate candidate materials in the lab. We solve this problem using the framework of Bayesian optimization (BO) and propose the first set of non-myopic methods for MOO problems. Prior work on non-myopic BO for single-objective problems relies on the Bellman optimality principle to handle the lookahead reasoning process. However, this principle does not hold for most MOO problems because the reward function needs to satisfy some conditions: scalar variable, monotonicity, and additivity. We address this challenge by using hypervolume improvement (HVI) as our scalarization approach, which allows us to use a lower-bound on the Bellman equation to approximate the finite-horizon using a batch expected hypervolume improvement (EHVI) acquisition function (AF) for MOO. Our formulation naturally allows us to use other improvement-based scalarizations and compare their efficacy to HVI. We derive three non-myopic AFs for MOBO: 1) the *Nested AF*, which is based on the exact computation of the lower bound, 2) the *Joint AF*, which is a lower bound on the nested AF, and 3) the *BINOM AF*, which is a fast and approximate variant based on batch multi-objective acquisition functions. Our experiments on multiple diverse real-world MO problems demonstrate that our non-myopic AFs substantially improve performance over the existing myopic AFs for MOBO.

## 1 Introduction

Many engineering design and scientific applications involve performing a sequence of experiments under a small-resource budget to optimize multiple *expensive-to-evaluate* objective functions. Consider the example of searching nanoporous materials optimized for hydrogen-powered vehicles. In this real-world problem, we need to perform physical lab experiments to make a candidate material and evaluate its volumetric and gravimetric hydrogen uptake properties. Our goal is to approximate the optimal Pareto front of solutions under a given *small resource budget* for experiments.

---

*\*These authors contributed equally to this work*

We consider the problem of finite-horizon sequential experimental design (SED) to solve multi-objective optimization (MOO) problems over expensive-to-evaluate objectives to find high-quality Pareto solutions. We solve this problem within the framework of Bayesian optimization (BO) (Shahriari et al., 2015), which has been shown to be sample-efficient in practice. The key idea behind BO is to train a surrogate model (e.g., a Gaussian process) from past experimental data, and intelligently select the future experiments guided by an acquisition function. Most of the prior acquisition functions for BO are myopic in nature (i.e., focused on the immediate advancement of the goal). There are a few non-myopic acquisition functions for BO in the single-objective setting (Jiang et al., 2020a; Lam & Willcox, 2016). These non-myopic strategies rely on formulating the optimal policy for SED as a dynamic program using the recursive Bellman optimality principle, and then solving for the optimal policy using methods with varying approximations (e.g., rollout strategies).

This work studies the design of non-myopic acquisition functions for MOO problems for the first time. The Bellman optimality principle does not hold in most MOO problems and requires the reward function of the underlying Markov decision process (MDP) to satisfy some conditions (Roijers et al., 2013; Van Moffaert & Nowé, 2014) 1) only scalar rewards; 2) monotonically increasing rewards with respect to better actions; and 3) additivity condition—for the optimality to hold. These conditions are challenging to satisfy in MOO problems and require a careful design of the reward function. We address this challenge by using hypervolume improvement (HVI) as our scalarization approach to define the reward function. Unlike the hypervolume function, hypervolume improvement satisfies the additivity condition. A key factor for these conditions to hold is to define optimality as Pareto optimality with respect to HVI rather than individual optimality for each objective function. This reward function allows us to use a lower-bound on the Bellman equation to approximately solve the finite-horizon SED problem using a batch utility function for MOO (e.g., qEHVI (Daulton et al., 2020)). Our approach generalizes to other scalarization approaches that satisfy the above conditions.

We derive three non-myopic multi-objective (NMMO) acquisition functions (AFs) for MOO with varying speed and accuracy trade-offs: 1) the *NMMO-Nested AF*, based on exact computation of the lower bound and computationally intractable even for small horizons, 2) the *NMMO-Joint AF*, a scalable lower bound on the nested AF, and 3) the *BINOM AF*, a fast and approximate version of EHVI. Our experiments on real-world MOO problems demonstrate that the proposed non-myopic acquisition functions substantially improve performance over the baseline BO methods for MOO.

**Contributions.** The key contribution of our work is the development and evaluation of principled non-myopic acquisition functions to solve MOO problems. In particular, we

- validate the use of a lower-bound on the Bellman optimality principle for MOO problems by using hypervolume improvement (HVI) to define an appropriate reward function;
- develop three non-myopic acquisition functions based on the derived lower bound—NMMO-Nested, NMMO-Joint, and BINOM—with varying speed and accuracy trade-offs;
- instantiate our framework with several information gain-based scalarizations and compare with HVI.
- experimentally evaluate the proposed non-myopic acquisition functions and baseline BO methods on multiple real-world MOO problems. Our code is provided at `https://github.com/Alaleh/NMMO`.

## 2 Problem Setup and Background

This section formally defines the problem setting and provides the background necessary for our solution approach.

**Multi-objective optimization (MOO).** Let $\mathfrak{X} \subset \mathbb{R}^d$ be the input space of $d$ design variables, where each candidate input $\mathbf{x} \in \mathfrak{X}$ is a $d$-dimensional vector. Let $\{f_1, \cdots, f_K\}$ with $K \geq 2$ represent the black-box objective functions defined over the input space $\mathfrak{X}$, where $f_1(\mathbf{x}), \cdots, f_K(\mathbf{x}) : \mathfrak{X} \to \mathbb{R}$. We denote the function evaluation at $\mathbf{x}$ as $\mathbf{y} = [y_1, \cdots, y_K]$, where $y_i = f_i(\mathbf{x})$ for all $i \in \{1, \cdots, K\}$. For example, to optimize nanoporous materials for hydrogen-powered vehicles, $\mathbf{x}$ represents a candidate material, $f_1(\mathbf{x})$ and $f_2(\mathbf{x})$

corresponds to a physical lab experiment to evaluate the volumetric and gravimetric hydrogen uptake of the material. WLOG, we assume maximization for all $K$ objective functions.

An input $\mathbf{x}$ Pareto-dominates another input $\mathbf{x}'$ if and only if $\forall i : f_i(\mathbf{x}) \geq f_i(\mathbf{x}')$ and $\exists j : f_j(\mathbf{x}) > f_j(\mathbf{x}')$. The optimal solution of the MOO problem is a set of inputs $\mathcal{X}^* \subset \mathfrak{X}$ such that no input $\mathbf{x} \in \mathfrak{X} \setminus \mathcal{X}^*$ Pareto-dominates another input $\mathbf{x}' \in \mathcal{X}^*$. The set of input solutions $\mathcal{X}^*$ is called the optimal *Pareto set*, and the corresponding set of function values $\mathcal{Y}^*$ is called the optimal *Pareto front*. In SED, we select one input for evaluation in each iteration, and our goal is to uncover a high-quality Pareto front while minimizing the total number of expensive function evaluations. A typical measure to evaluate the quality of a Pareto front is the *Pareto hypervolume (PHV)* indicator (Zitzler & Thiele, 1999).

**Pareto Hypervolume (HV) Indicator.** The hypervolume indicator $HV$ of a finite approximate Pareto set $\mathcal{Y} \subset \mathbb{R}^K$ is the $K$-dimensional Lebesgue measure $\lambda_K$ of the space dominated by $\mathcal{Y}$ and bounded from below by a given reference point $\mathbf{r} \in \mathbb{R}^K$:

$$HV(\mathcal{Y}, \mathbf{r}) = \lambda_K \bigg( \bigcup_{\mathbf{y} \in \mathcal{Y}} [\mathbf{r}, \mathbf{y}] \bigg), \tag{1}$$

where $[\mathbf{r}, \mathbf{y}]$ denotes the hyper-rectangle formed by the reference point $\mathbf{r}$ and the outcome vector $\mathbf{y}$. The hypervolume quantifies the quality of the Pareto front approximation, with larger values indicating better solutions. In SED, we aim to select inputs that maximize the increase in hypervolume to efficiently improve the Pareto front.

**Non-myopic MOO problem.** Our goal is to approximate the optimal Pareto front for a MOO problem within a maximum experimental budget $T$, which is typically small (i.e., finite-horizon SED). This problem can be formulated as:

$$\max_{X \in P(\mathfrak{X})} \max_{\mathbf{x} \in X} [f_1(\mathbf{x}), f_2(\mathbf{x}) \cdots, f_K(\mathbf{x})], \qquad \text{s.t. } |X| = T, \tag{2}$$

where $P(\mathfrak{X})$ denotes the power set of $\mathfrak{X}$ and $X = \{\mathbf{x}_1 \ldots \mathbf{x}_T\}$ is the set of inputs selected for function evaluation. We solve this problem using a non-myopic decision policy, where, at each iteration, the algorithm selects one input for evaluation while reasoning about candidate experimental design plans within the budget $T$.

**Overview of Bayesian optimization (BO).** BO is an effective framework for solving expensive black-box optimization problems. BO has three main steps: (i) conducting an expensive experiment with an input selected by BO, (ii) updating the surrogate model by including the new evaluations, and (iii) selecting a new candidate input for the next function evaluation. The two key components of BO are the surrogate model and an acquisition function. Surrogate models are probabilistic models (typically Gaussian processes (Williams & Rasmussen, 2006)) that are trained on data from past function evaluations. A surrogate model is able to predict the output of unknown inputs without requiring an expensive experiment and to quantify the uncertainty of its predictions. The acquisition function (e.g., expected improvement (EI) (Mockus, 1989)) is used to decide which input to select for the next function evaluation. It uses the surrogate model to score the utility of selecting inputs for the next experiment by trading off exploitation and exploration. The goal is to quickly direct the search toward high-quality solutions with the least number of iterations. BO in a multi-objective setting requires particular consideration for modeling and acquisition function design. For MOO problems, we model the objective functions $f_1, \cdots, f_K$ using $K$ independent GP models $\mathcal{GP}_1, \cdots, \mathcal{GP}_K$ and discuss acquisition functions in Section 3.

**Non-myopic decision policies for BO.** We review important facts for optimal sequential decision-making (Osborne, 2010). Consider collecting $t$ observations $D_t = \{(\mathbf{x}_i, \mathbf{y}_i), i \in [0, t]\}$, and let $u$ denote a utility function defining a quality metric over the data $D_t$. The marginal gain in the utility of query $\mathbf{x}$ w.r.t. $D_t$ is expressed as

$$u(\mathbf{y}|\mathbf{x}, D_t) = u(D_t \cup (\mathbf{x}, \mathbf{y})) - u(D_t). \tag{3}$$

Consider the case where $T - t$ steps are remaining. The expected marginal gain for $T - t$-steps is expressed through the Bellman recursion (Jiang et al., 2020a) as

$$\mathcal{U}_{T-t}(\mathbf{x}|D_t) = \underbrace{\mathbb{E}_{\mathbf{y}}[u(\mathbf{y}|\mathbf{x}, D_t)]}_{\text{exploitation}} + \underbrace{\mathbb{E}_{\mathbf{y}}\left[\max_{\mathbf{x}'}\mathcal{U}_{T-t-1}(\mathbf{x}'|D_t \cup (\mathbf{x}, \mathbf{y}))\right]}_{\text{exploration}}. \tag{4}$$

The optimal expected policy to select the $T - t$ lookahead steps can be obtained by sequentially maximizing Equation 4 at each step

$$\mathbf{x}^* = \underset{\mathbf{x} \in \mathfrak{X}}{\operatorname{argmax}} \ \mathcal{U}_{T-t}(\mathbf{x}|D_t). \tag{5}$$

Being a sequence of $T - t$ nested integrals, solving the optimization problem in equation 5 to optimality is intractable for even a small value of $T - t$ (Lam et al., 2016; Jiang et al., 2020b).

Recent work made a connection between non-myopic input selection and batch input selection (Jiang et al., 2017; 2020a). Assuming parallel evaluation, the expected marginal utility of a new batch of evaluations $Y = \{\mathbf{y}_t, \cdots, \mathbf{y}_T\}$ given their associated batch of inputs $X = \{\mathbf{x}_t, \cdots, \mathbf{x}_T\}$ is defined as:

$$\mathcal{U}(X|D_t) = \mathbb{E}_Y[u(Y|X, D_t)] \tag{6}$$

Respectively, the maximum expected marginal utility of a batch of evaluations at input locations $X'$ of size $|X'| = T - t - 1$ conditioned on the addition of an observation $(\mathbf{x}, \mathbf{y})$ to the data is defined as

$$\max_{X':|X'|=T-t-1} \mathbb{E}_{\mathbf{y}}[\mathcal{U}(X'|D_t \cup (\mathbf{x}, \mathbf{y}))]. \tag{7}$$

Comparing equations 4 and 7, the second term in equation 4 has an adaptive utility with a nested maximization while equation 7 has a joint batch utility with an outer maximization. This leads to a lower-bound connection between the two:

$$\max_{X':|X'|=T-t-1} \mathbb{E}_{\mathbf{y}}[\mathcal{U}(X'|D_t \cup (\mathbf{x}, \mathbf{y}))] \leq \mathbb{E}_{\mathbf{y}}[\max_{\mathbf{x}'}\mathcal{U}_{T-t-1}(\mathbf{x}'|D_t \cup (\mathbf{x}, \mathbf{y}))]. \tag{8}$$

Consequently, we obtain a lower bound on the expected marginal gain defined by the Bellman recursion in equation 4

$$\alpha(\mathbf{x}|D_t) \leq \mathcal{U}_{T-t}(\mathbf{x}|D_t) \tag{9}$$

$$\text{with} \quad \alpha(\mathbf{x}|D_t) = \underbrace{\mathbb{E}_{\mathbf{y}}[u(\mathbf{y}|\mathbf{x}, D_t)]}_{exploitation} + \underbrace{\max_{X':|X'|=T-t-1} \mathbb{E}_{\mathbf{y}}[\mathcal{U}(X'|D_t \cup (\mathbf{x}, \mathbf{y}))]}_{exploration}. \tag{10}$$

The lower bound on the Bellman equation (Equation 10) provides an efficient alternative for optimizing the sequential decision-making process in non-myopic single-objective BO (Jiang et al., 2020a), where the utility function $u(D_t)$ is defined as the incumbent, and the marginal utility $u(y|\mathbf{x}, D_t)$ as the incumbent improvement. However, applying the same strategy in the multi-objective BO (MOBO) setting is non-trivial. The Bellman equation is inherently single-dimensional with respect to the output of the utility function and, therefore, presents challenges when faced with the multi-output nature of MOBO, where conflicting objectives require simultaneous consideration. This necessitates an adapted formulation capable of capturing the trade-offs across multiple objectives and guiding the exploration and exploitation balance. The methodological challenges to address the extension of the Bellman equation for MOBO and the proposed solutions are described in Section 4.

## 3 Related Work

Most of the prior work on acquisition functions in BO falls under the myopic category. There is less work on non-myopic acquisition functions with a focus on single-objective BO.

**Myopic single-objective BO.** Recent advances in single-objective BO have predominantly focused on refining and extending traditional acquisition functions. The most notable techniques (Garnett, 2023; Wang & Jegelka, 2017; Hernández-Lobato et al., 2014; Ament et al., 2024) use Gaussian processes as surrogate models.

**Non-myopic single-objective BO.** Non-myopic approaches in single-objective BO (Jiang et al., 2020a; Lam & Willcox, 2016; González et al., 2016; Ginsbourger & Le Riche, 2009; Osborne, 2010; Osborne et al., 2009; Belakaria et al., 2022; Kharkovskii et al., 2020; Marchant et al., 2014; Ling et al., 2016) represent an emerging direction aimed at overcoming the limitations of myopic acquisition functions, which typically optimize for immediate gains. These lookahead methods consider the future impact of current decisions (Wu & Frazier, 2019; Lee et al., 2021), thereby enhancing long-term optimization outcomes when the experimental resource budget is small. Jiang et al. (2017) also addressed non-myopic active search by introducing batch selection to approximate non-myopic sequential decision-making.

**Multi-objective BO.** When compared to single-objective BO, multi-objective BO (MOBO) has received relatively less attention. The predominant methods for designing MOBO strategies include information-theoretic methods (Tu et al., 2022; Hernández-Lobato et al., 2016; Belakaria et al., 2019; Suzuki et al., 2020; Garrido-Merchán & Hernández-Lobato, 2021), uncertainty based methods (Belakaria et al., 2020b), hypervolume-based methods (Konakovic Lukovic et al., 2020; Daulton et al., 2020; Ahmadianshalchi et al., 2024a), scalarization-based methods (Daulton et al., 2022; Knowles, 2006), and batch MOBO methods (Ahmadianshalchi et al., 2024b). The joint entropy search for MOBO (JESMO) algorithm (Tu et al., 2022) aims to maximize information gain about the optimal Pareto front by evaluating inputs that maximally reduce its uncertainty. The expected hypervolume improvement (EHVI) method (Daulton et al., 2020) extends the concept of EI from a single-objective to a MOO problem by assessing the potential improvement in hypervolume from new input evaluations. Several efforts to extend these approaches to handle constraints and preferences have resulted in more general frameworks that accommodate complex real-world decision-making scenarios (Garrido-Merchán & Hernández-Lobato, 2019; Belakaria et al., 2020a; Ahmadianshalchi et al., 2024c). MOBO methods have been applied to solve challenging engineering design applications (Chen et al., 2024; Belakaria et al., 2020c; Deshwal et al., 2021a; Yang et al., 2021; Zhou et al., 2020) and can also be employed with surrogate models for combinatorial spaces (Deshwal & Doppa, 2021; Deshwal et al., 2023; 2021b; 2022).

**Non-myopic MOBO.** Despite advances in myopic strategies for MOBO, a notable knowledge gap remains. There is no prior work on non-myopic acquisition functions for MOBO problems with a finite horizon. A recent approach proposed for settings with decoupled function evaluation, Daulton et al. (2023) considers a one-step lookahead only and thus lacks strategic foresight. *Our goal is to precisely fill this critical gap in the current state of knowledge.*

## 4 Non-Myopic Multi-Objective BO

In this section, we provide details of our proposed approach for non-myopic MOBO. We first discuss the challenge of maintaining the validity of the Bellman optimality principle and, by consequence, the ability to use the Bellman equation and its lower-bound to solve the SED problem in the multi-objective setting. We then discuss design choices for the marginal utility function that enables the effective use of the Bellman equation in our setting. We finally provide the details of the non-myopic acquisition functions we propose, their computational challenges, and practical algorithms.

### 4.1 Challenges of Non-Myopic MOBO

For the Bellman optimality principle to hold and for the optimal policy to be computable by solving the Bellman equation, the reward function has to satisfy several criteria. In our problem setting, we refer to the reward and the marginal gain in utility defined in equation 3 interchangeably. We state below some of the reward function requirements that are not straightforward to satisfy when dealing with a multi-objective problem:

- *Scalarization:* the reward has to be defined as a single scalar value and not a vector of values (Boutilier et al., 1999; Roijers et al., 2013). In single-objective optimization, the reward is defined as the improvement in the best-found function value (Lam et al., 2016; Jiang et al., 2020a). However, in MOBO problems, every input evaluation leads to a vector of values representing a trade-off where some of the objectives may improve while others degrade.
- *Monotonicity:* the reward has to be a monotonically increasing function with respect to the quality of the actions (i.e., better actions lead to higher rewards). This monotonic property ensures that the optimization trajectory is aligned with improving performance (Roijers et al., 2013).
- *Additivity:* the reward must satisfy the additivity condition, i.e, the total reward is a sum of the rewards obtained at each step of the decision-making process (Boutilier et al., 1999; Roijers et al., 2013; Van Moffaert & Nowé, 2014). Additivity is essential for the recursive decomposition of the sequential decision process.

It is important to note that the choice of the scalarization approach plays a key role in preserving the monotonicity and additivity conditions. In MOBO problems, the challenge lies in combining both the improvement and degradation incurred in several objectives into a single, quantifiable metric that accurately reflects the quality of an input in terms of Pareto dominance regardless of trade-offs in individual objectives. In broader applications of the Bellman equation (e.g., multi-objective reinforcement learning), linear scalarization is the most common technique known to preserve the additivity of the reward, where each objective is multiplied by a predetermined nonnegative weight reflecting its relative importance (Barrett & Narayanan, 2008). Since linear scalarization computes a convex combination, it has the known fundamental limitation that it cannot recover solutions laying outside the convex regions of the Pareto front (Van Moffaert & Nowé, 2014).

### 4.2 Our Proposed Non-Myopic MOBO Method

In this section, we provide the details of the design choices of our utility function that enable the application of the Bellman equation in the MO setting. We then provide the details of the different acquisition functions we propose for non-myopic MOBO.

#### 4.2.1 Utility for Multi-Objective Non-Myopic Setting

The hypervolume (HV) quality indicator provides a scalar measure of a particular Pareto front $\mathcal{Y}$. HV is the volume between a reference point and the Pareto front. This measure has been of particular interest in MOBO problems because it is known to be strictly monotonically increasing with regard to Pareto dominance. However, the HV measure does not satisfy the additivity condition necessary for Bellman's optimality principle (Van Moffaert et al., 2013). In this work, we propose to use the *hypervolume improvement* (HVI) to scalarize our multiple single-objective rewards defined as the *improvement* of each of the functions. This approach quantifies the quality of a new point $\mathbf{y}$ by the volume of the objective space that is newly encompassed by extending the Pareto front to include the new point.

$$HVI(\mathbf{y}|\mathcal{Y}) = HV(\mathcal{Y} \cup \mathbf{y}) - HV(\mathcal{Y}) \tag{11}$$

HVI provides a suitable scalarization because it naturally satisfies all of the reward requirements. First, it integrates the improvement of multiple objectives into a single scalar value. Second, it preserves the monotonicity by reflecting the *Pareto quality* of a new input (action) given the current Pareto front (state space representation).

Intuitively, HVI satisfies the additivity condition because the improvement in hypervolume by successive input evaluations (actions) can be summed to give a total improvement over the sequence of evaluations, thus adhering to the required recursive structure.

**Lemma 1.** *We denote $HVI_t$ the hypervolume improvement at $t$ with $t \in [1, T]$ and $HVI_{total}$ be total hypervolume improvement collected at iteration $T$. The additivity condition is satisfied because the total hypervolume collected at iteration $T$ can be decomposed as a summation of the intermediate hypervolume improvements*

$$HVI_{total} = \sum_{t=1}^{t=T} HVI_t \tag{12}$$

We provide a proof of lemma 1 in Appendix B. It is important to note that the Bellman optimality principle holds with respect to HVI and Pareto optimality defined by the HV measure, rather than with respect to each objective function separately. This definition of optimality aligns well with the goals of MOBO, where the focus is on maximizing the objective space coverage efficiently, and where defining optimality with respect to HV and using HV as the primary evaluation metric is the most common approach (Belakaria et al., 2019; Daulton et al., 2021). We make this distinction to avoid confusion with the multi-objective RL setting, where theoretical optimality guarantees with respect to each function independently might not hold when using nonlinear scalarizations (Roijers et al., 2013).

### 4.2.2 Proposed Acquisition Functions

Given our choice of reward as the HVI, we can define our SED problem using the Bellman equation. We formally define the marginal gain in the utility of our multi-objective setting as follows:

$$u(\mathbf{y}|\mathbf{x}, D_t) = HVI(\mathbf{y}|\mathcal{Y}). \tag{13}$$

Similar to the single-objective case, computing the optimal policy by fully solving the Bellman equation to optimality is intractable even for a moderately large horizon $T - t$ (Jiang et al., 2020b). An effective approach to navigating this complexity is to approximate the optimal policy by optimizing the lower-bound on the Bellman equation (Equation 9). In what follows, we denote by $X$ the full remaining horizon at iteration $t$ with $|X| = T - t$, $\mathbf{x}$ the next input to evaluate and the first input in the horizon $X$, and $X'$ the lookahead horizon with $|X'| = T - t - 1$ and $X = \{\mathbf{x}, X'\}$. The selection of the next input to evaluate while accounting for the lookahead horizon (the remaining sequence of inputs in the horizon) can be achieved by maximizing the acquisition function $\alpha(\mathbf{x}|D_t)$ (Equation 10). By defining the marginal gain in utility as the HVI, we rewrite the acquisition function as:

$$\alpha_{Nested}(\mathbf{x}|D_t) = EHVI(\mathbf{x}|D_t) + \max_{X':|X'|=T-t-1} \mathbb{E}_{\mathbf{y}}[BEHVI(X'|D_t \cup (\mathbf{x}, \mathbf{y}))] \tag{14}$$

The first term on the r.h.s is the expected hypervolume improvement (EHVI) for the input $\mathbf{x}$ given the observed data $D_t$. The second term is the maximum of the expectation over the batch expected hypervolume improvement (BEHVI) computed at the batch of inputs $X'$ conditioned on the addition of the new observation $(\mathbf{x}, \mathbf{y})$ to the data. The expectation is computed with respect to the possible values of $\mathbf{y}$ sampled from the posterior of the surrogate models at the input $\mathbf{x}$. We refer to this new acquisition function as the *nested non-myopic multi-object acquisition function*. Since the first part corresponds to the EHVI acquisition function (Emmerich & Klinkenberg, 2008), which corresponds to the one-step myopic policy, input selection by optimizing this lower bound is always at least as tight as optimizing the myopic policy. This approach, while approximate, leverages the strengths of the EHVI and BEHVI acquisition functions (Emmerich & Klinkenberg, 2008; Daulton et al., 2020) to efficiently compute the immediate impact of the selection of input $\mathbf{x}$ and to approximate the impact of the selection of $\mathbf{x}$ on the lookahead horizon $X'$.

---

**Algorithm 1** NMMO Algorithm

---

**Input**: input space $\mathfrak{X}$; $K$ blackbox objective functions $f_1(x), f_2(x), \cdots, f_K(x)$; and maximum no. of iterations $T$, selected non-myopic method $\in$ {NMMO-Joint, NMMO-Nested, BINOM}.

1: Initialize GP models $\mathcal{GP}_1, \mathcal{GP}_2, \cdots, \mathcal{GP}_K$ by evaluating at $N_0$ initial points
2: **for** each iteration $t = 1$ to $T$ **do**
3:     **if** method==NMMO-Nested **then**
4:         Select $\mathbf{x}_t \leftarrow \arg max_{\mathbf{x} \in \mathfrak{X}} \ \alpha_{Nested}(\mathbf{x}|D_t)$, where $\alpha_{Nested}$ is defined in Equation 14
5:     **if** method==NMMO-Joint **then**
6:         Select $\mathbf{x}_t, X' \leftarrow \arg max_{\mathbf{x} \in \mathfrak{X}, X' \in \mathfrak{X}^{T-t-1}} \ \alpha_{Joint}(\mathbf{x}, X'|D_t)$, where $\alpha_{Joint}$ is defined in Equation 15
7:     **if** method==BINOM **then**
8:         Select $X \leftarrow \arg max_{X \in \mathfrak{X}^{T-t}} \ \alpha_{BINOM}(X|D_t)$, where $\alpha_{BINOM}$ is defined in Equation 16
9:         Select $\mathbf{x}_t \leftarrow \arg max_{\mathbf{x} \in X} \ EHVI(\mathbf{x}|D_t)$
10:    Evaluate $\mathbf{x}_t$: $\mathbf{y}_t \leftarrow (f_1(\mathbf{x}_t), \cdots, f_K(\mathbf{x}_t))$
11:    Aggregate data: $D_{t+1} \leftarrow D_t \cup \{(\mathbf{x}_t, \mathbf{y}_t)\}$
12:    Update models $\mathcal{GP}_1, \mathcal{GP}_2, \cdots, \mathcal{GP}_K$
13: **end for**
14: **return** Pareto front of $f_1(x), \cdots, f_K(x)$ based on $D_T$

---

**Computational challenges.** The expression of the acquisition function (Equation 14) involves two computational challenges for which we propose solutions.

*1. Nested Maximization:* The second term in the nested non-myopic AF for MOO (Equation 14) is defined as a maximization problem. Thus, evaluating the acquisition function for each candidate input $\mathbf{x}$ requires solving an internal optimization problem to estimate the batch representing the lookahead horizon. This leads to a computationally prohibitive acquisition function optimization. To address this challenge, we propose two alternative acquisition functions:

**Alternative lower-bound computation**: One approach to circumvent the nested maximization problem is to reformulate the acquisition function as a new lower-bound that jointly optimizes over both the first input in the horizon $\mathbf{x}$ and the lookahead horizon $X'$. We refer to this approach as the *joint* non-myopic MO acquisition function.

$$\begin{aligned} \alpha_{Joint}(\mathbf{x}, X'|D_t) &= \mathbb{E}_{\mathbf{y}}[u(\mathbf{y}|\mathbf{x}, D_t)] + \mathbb{E}_{\mathbf{y}}[\mathcal{U}(X'|D_t \cup (\mathbf{x}, \mathbf{y}))] \\ &= EHVI(\mathbf{x}|D_t) + \mathbb{E}_{\mathbf{y}}[BEHVI(X'|D_t \cup (\mathbf{x}, \mathbf{y}))] \end{aligned} \tag{15}$$

**Approximation via batch selection**: Approximating the selection of all the inputs in the horizon by a batch acquisition function has been previously used for non-myopic single-objective BO (González et al., 2016; Jiang et al., 2020a). Let $X^*$, s.t $|X^*| = T - t$ be a set of inputs selected by maximizing a batch acquisition function $\mathcal{U}(X|D_t)$. Jiang et al. (2020a) showed that choosing any input $\mathbf{x}^* \in X^*$ can be approximated by selecting inputs that maximize $\alpha(\mathbf{x}|D_t)$ several times, with $\alpha$ as the acquisition function in equation 10. With this approximation, we propose to follow Jiang et al. (2020a), and approximate the non-myopic acquisition function by a fully joint batch acquisition function. We select the full horizon by maximizing the BEHVI. We call this selection approach Batch-Informed NOnmyopic Multi-objective optimization (BINOM):

$$\alpha_{BINOM}(X|D_t) = \mathcal{U}(X|D_t) = BEHVI(X|D_t). \tag{16}$$

However, an important issue arises. The strategy does not differentiate the next input for evaluation $\mathbf{x}^*$ from the lookahead horizon $X'$ in the way the acquisition functions in equations 10 and 15 do. By analogy to Jiang et al. (2020a), we select the next input to evaluate as the input with the highest immediate EHVI among the inputs in the selected batch.

*2. Expectation computation via posterior sampling:* The second term of the acquisition function (Equation 10) requires an expectation computed over the BEHVI. This expectation can be estimated by sampling several realizations of **y** from the posterior of the surrogate models. The accuracy of the expectation estimation depends on the number of samples used. In practice, more samples lead to a slower and more computationally expensive acquisition function, while fewer samples leads to inaccurate estimation and a higher variance. This results in unstable optimization behavior with unreliable outcomes. To mitigate the former issues, we propose to substitute the expectation by the use of a one-step lookahead GP model (Lyu et al., 2021). In this approach, the mean function of the GP remains unchanged, while the variance is updated based on the posterior conditioning on the newly added input.

In summary, these computational strategies are crucial for enhancing the efficiency and stability of the acquisition functions in multi-objective optimization scenarios. With these strategies, our decision-making process for finite-horizon sequential experimental design using MOBO becomes computationally tractable and more robust.

## 5 Experiments and Results

This section describes our experimental evaluation comparing the proposed non-myopic methods with baselines on several real-world and synthetic MOO problems.

**Benchmark MOO problems:** We provide a brief description of our real-world MOO benchmarks, including the number of input dimensions ($d$) and the number of objectives ($K$).

*Metal-organic Framework Design ($d = 7$, $K = 2$)* (Kitagawa et al., 2014; Boyd et al., 2019): Nanoporous materials (NPMs) (Lu & Zhao, 2004) represent a class of structures characterized by their microscopic pores at the nanoscale level. Metal-organic frameworks (MOFs) (Ahmed et al., 2017; Ahmed & Siegel, 2021; Boyd et al., 2019; Deshwal et al., 2021c) are a class of NPMs, instrumental in several sustainable applications due to their selective gas absorption capabilities. The goal of this problem is to identify the Pareto front of MOFs from $\approx$ 100K candidates that optimize gravimetric and volumetric hydrogen uptake for hydrogen-powered vehicles.

*Reinforced Concrete Beam Design ($d = 3$, $K = 2$)* (Amir & Hasegawa, 1989): The goal is to optimize the cost of concrete and reinforcing steel used in the construction of a beam. Its input features are the area of reinforcement, the beam's width, and its depth. This setup enabled us to explore a mixed input space to find cost-effective yet robust designs.

*Four-Bar Truss Design ($d = 4$, $K = 2$)* (Cheng & Li, 1999): The goal is to minimize the structural volume and joint displacement of a four-bar truss to reduce the weight of the structure and the displacement of a specific node. The features are the areas of the four-member cross-sections.

*Gear Train Design ($d = 4$, $K = 3$)* (Deb & Srinivasan, 2006; Tanabe & Ishibuchi, 2020): The goal is to design a compound four-gear train to achieve a specific target gear ratio between the driver and driven shafts. The objectives are to minimize the error between the obtained gear ratio and the desired gear ratio and to minimize the maximum size of the four gears to optimize the compactness of the design.

*Welded Beam Design ($d = 4$, $K = 3$)* (Coello & Montes, 2002; Rao, 2019): The goal is to minimize the cost of fabrication and the end deflection of a welded beam by precisely adjusting four of its side lengths. To focus more directly on the relationship between the beam's dimensions and the optimization objectives, we adopted a non-constrained version of this problem as in prior work (Tanabe & Ishibuchi, 2020; Konakovic Lukovic et al., 2020).

*Disc Break Design ($d = 4$, $K = 3$)* (Ray & Liew, 2002; Tanabe & Ishibuchi, 2020): The goals are to minimize the mass of the brake and the stopping time. The variables are the inner and outer radius of the discs, the engaging force, and the number of friction surfaces.

We also include the ZDT-3 ($d = 9$, $K = 2$) (Zitzler et al., 2000) problem as a synthetic MOO benchmark. We provide additional results in the Appendix (Section A.4) and an overview of all our MOO benchmarks(Appendix Table 1).

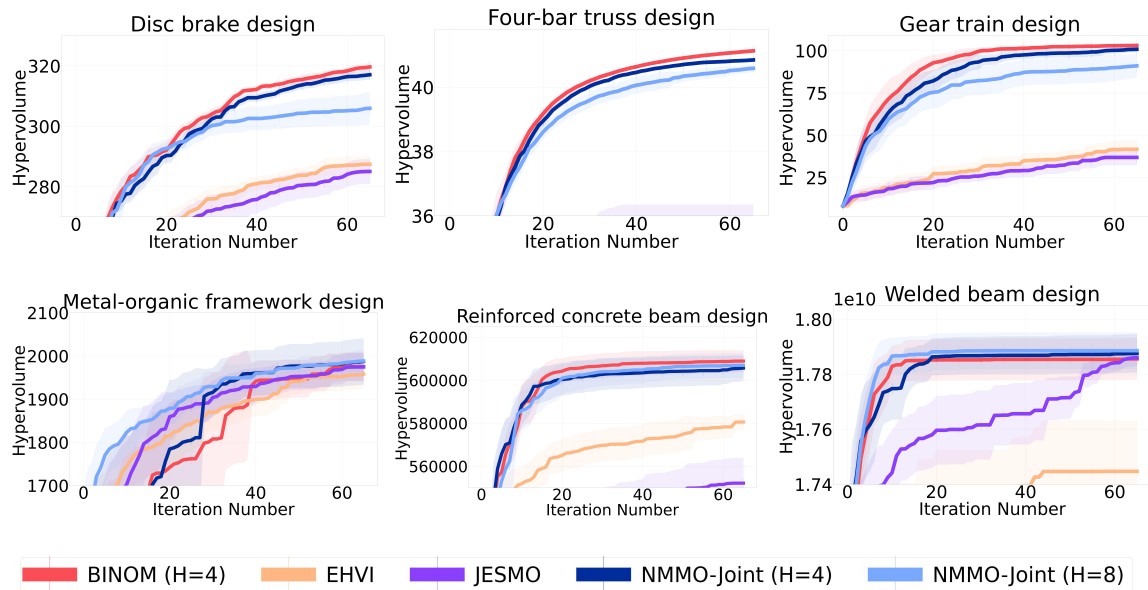

Figure 1: Hypervolume results on real-world problems: Non-myopic methods vs. EHVI and JESMO.

**Baselines:** We proposed three non-myopic acquisition functions (NMMO-Joint, NMMO-Nested, and BINOM) with varying trade-offs between computational complexity and approximation accuracy. In an ideal scenario with a perfect surrogate model, evaluating the entire lookahead horizon $H = T - t$ at each iteration $t$ would yield the closest results to the optimal policy. However, previous research on non-myopic Bayesian optimization (Jiang et al., 2020a;b; Lee et al., 2021; 2020) has shown that models often struggle with long-term predictions due to compounding uncertainty at each lookahead step. This is because the model's predictions at each step are based on the uncertain predictions from the previous steps. As a result, querying an extended horizon can hinder the optimization process by including suboptimal or misleading inputs in the non-myopic horizon, potentially guiding the search in less promising directions and incurring higher computational costs. To address this issue, we adopt the same strategy from previous work and introduce a maximum horizon length to limit the size of the horizon.

The NMMO-Nested method stands out with its potential for superior optimization performance due to its comprehensive approach to handling nested optimizations. However, this method has practical challenges. Notably, this method requires a pre-defined grid for optimization, as internal maximization prevents using gradient-based optimizers. Furthermore, Its computational complexity constrains the size of the grid that can be feasibly used, which in turn limits the method's operational speed and accuracy. Hence, the nested approach does not perform as well as expected compared to the other proposed non-myopic approaches. We provide results including NMMO-Joint with lookahead horizon $H \in \{4, 8\}$ and BINOM with $H = 4$. We report the results for NMMO-Nested with $H = 2$ in the Appendix Section A.5.

We consider two state-of-the-art MOBO baselines to benchmark the performance of our non-myopic methods. First, we use the multi-objective variant of the joint entropy search for multi-objective Bayesian optimization (JESMO) (Tu et al., 2022; Balandat et al., 2020). Second, we use the expected hypervolume improvement (EHVI) method (Emmerich et al., 2006; Balandat et al., 2020) as a hypervolume-based baseline. Another baseline is the hypervolume knowledge gradient (HVKG), a one-step lookahead method for cost-aware MOBO problems (Daulton et al., 2023). While the HVKG algorithm is designed for cost-aware settings with decoupled evaluations, we use a modified version of this method where the objective function costs are set to zero, and we calculate all objectives for a true evaluation of the hypervolume. We report the results in the Appendix Section A.5.

The baseline methods were chosen due to their distinct but complementary approaches to handling multi-objective trade-offs, providing a comprehensive comparative perspective against our proposed NMMO method. The effectiveness of each method is measured using the hypervolume indicator.

**Evaluation metric.** The Hypervolume is a widely-used metric in MOO that measures the volume of the space dominated by the Pareto front relative to a predefined reference point: a larger hypervolume indicates better Pareto solutions. We will compare the hypervolume metric after each BO iteration.

**Experimental details:** All experiments were averaged among 15 runs, initialized with five points randomly generated from Sobol sequences, and run for 65 iterations. We emphasize that non-myopic solutions are well-suited for the smaller experimental budgets. We model each of the multiple objectives using an independent GP with a Matérn 5/2 ARD kernel. We use implementations from the BoTorch Python package (Balandat et al., 2020) for all baselines, including JESMO, EHVI, and HVKG. All methods were evaluated under identical initial conditions. This includes consistent initialization, identical computational budgets, and the same evaluation criteria across all experiments.

### 5.1   Results and Discussion

Figure 1 shows the hypervolume versus BO iterations for NMMO-Joint with lookahead horizon $H \in \{4, 8\}$ and BINOM with lookahead horizon $H = 4$ against existing state-of-the-art baselines.

**Non-myopic methods perform better than myopic ones.** The non-myopic methods outperform the state-of-the-art myopic MOBO algorithms on all six real-world MOO problems (Figure 1). These results highlight the capability of non-myopic methods in maximizing the hypervolume when the experimental budget is small (i.e., 65 BO iterations).

**Non-myopic methods with different horizons.** From our results, we often observe a slight decrease in performance with increasing horizon values in non-myopic MOO. This trend can be primarily attributed to the increased complexity and model uncertainty as the decision horizon extends. A longer horizon necessitates the consideration of a larger set of future outcomes and their interactions, which not only complicates the optimization task but also introduces greater uncertainty and potential for errors in prediction. Consequently, these accumulated errors can degrade the optimization strategy, making it challenging for the algorithm to effectively balance short-term gains against long-term goals, and negatively impacting performance. This finding highlights the challenges in implementing long-term strategic planning in complex optimization environments. These challenges can be seen in the NMMO-Joint and BINOM methods run on each benchmark with increasing lookahead horizon $H \in \{2, 4, 6, 8\}$ (Appendix Figure 2 and Figure 3).

**Computational runtime comparison.** In our comparative analysis of myopic and non-myopic MOO methods, there are computational complexity and optimization performance trade-offs. Myopic MOO methods, due to their focus on immediate gains, have lower computational complexity, which translates into faster runtimes (Table 2). However, this computational efficiency comes at the cost of performance. Our results indicate that, while myopic methods are faster, they underperform in comparison to their non-myopic counterparts in limited-budget settings. Hence, while non-myopic approaches require more computational resources, their better optimization results justify the additional complexity.

**Ablation study of information-gain within BINOM.** In addition to the EHVI acquisition function, we incorporated information-gain acquisition functions, specifically MESMO and JESMO, into our proposed BINOM approach. Even though information gain based scalarization does not always satisfy the monotonicity condition as discussed in section A.3, These instantiations of our framework provide a complementary approach to the hypervolume-based strategies. The results, provided in the Appendix demonstrate that the non-myopic methods maintain strong performance across different instantiations of BINOM, highlighting the versatility and effectiveness of our framework. The results also show that the HVI instantiation performs better in most cases.

# 6 Summary

This paper introduced a novel set of non-myopic methods for multi-objective Bayesian optimization (MOBO), to specifically address the problem of finite-horizon sequential experimental design (SED) for MOO problems. By addressing the difficulties related to scalarization, monotonicity, and additivity of the reward function typically encountered in MOO scenarios, we successfully adapted the Bellman optimality principle for MOBO by adopting a new approach to scalarization using hypervolume improvement (HVI). Our proposed methods, including the Nested, Joint, and BINOM acquisition functions, represent the first non-myopic acquisition functions tailored for MOBO problems. Our experiments on multiple real-world problems, show that our methods show significant improvements by consistently outperforming the state-of-the-art myopic approaches in limited-budget settings.

**Acknowledgments.** Belakaria is supported by a Stanford Data Science Postdoctoral Fellowship. Ahmadianshalchi and Doppa are supported by the NSF CAREER award IIS-1845922.

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

# A  Appendix

## A.1  Benchmarks details

In Table 1, we include a concise overview of all of our MOO benchmarks consisting of the problem names, number of input dimensions ($d$), number of output dimensions ($K$), and the reference point ($r$) used for calculating the hypervolume indicator for each benchmark.

## A.2  Analysis of lookahead horizons

We include figures illustrating the performance of our non-myopic methods across different horizon values. The results reveal a general trend where larger horizons correlate with slight drops in performance. This observation can primarily be attributed to the increased uncertainty associated with extending the optimization horizon. As the horizon increases, the predictive model must account for a broader range of potential future outcomes. Their uncertainties can complicate the decision-making process, as the model faces increased difficulty in accurately estimating the long-term consequences of current decisions. Consequently, while longer horizons theoretically allow for more strategic and informed decision-making by considering future impacts, the added uncertainty can undermine performance by introducing greater variability in the optimization outcomes. Figure 2 and Figure 3 show the hypervolume results for NMMO-Joint and BINOM with horizon lengths 2, 4, 6, and 8.

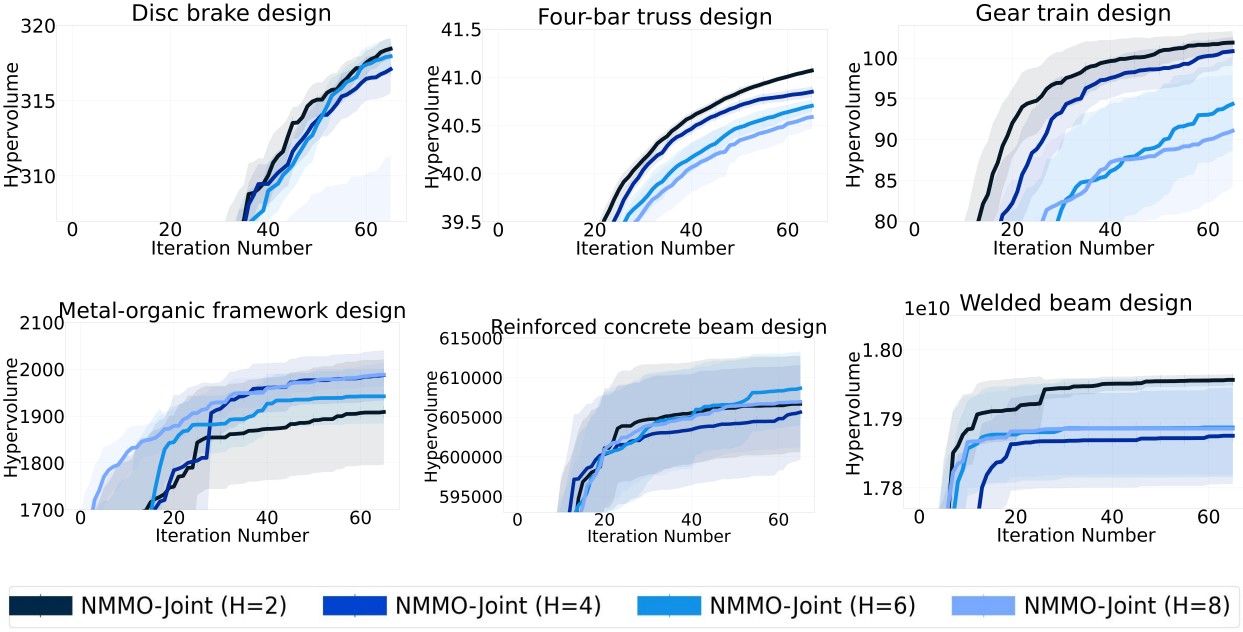

Figure 2: Hypervolume results for NMMO-Joint with lookahead horizon $H \in \{2, 4, 6, 8\}$.

## A.3  Alternative scalarization approaches: an ablation study

The proof of additivity in Lemma 1 relies on the fact that HVI is an improvement-based scalarization. Therefore, this lemma theoretically holds for any improvement-based scalarization. Notably, information gain (IG) based scalarizations, defined as the decrease in the entropy of the Pareto front/set, can indeed satisfy the additivity condition. IG can be computed with respect to the input space (leading to an instantiation of our method with PESMO (Hernández-Lobato et al., 2016)), the output space (leading to an instantiation of our method with MESMO (Belakaria et al., 2019; 2021)), or both (leading to an instantiation of our method with JESMO (Tu et al., 2022)). However, it is important to point out that the IG monotonicity with

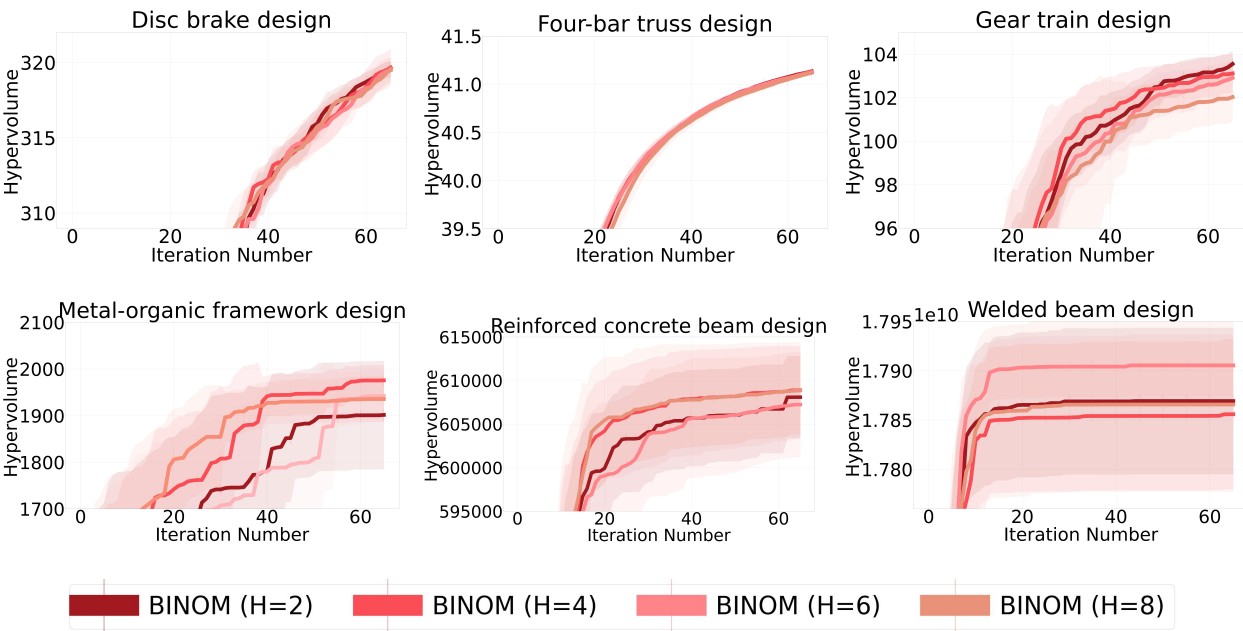

Figure 3: Hypervolume results for BINOM with lookahead horizon $H \in \{2, 4, 6, 8\}$.

respect to better actions might not always hold. Typically, an input is considered a better choice/action, if it dominates existing points in the Pareto front and thus enhances the quality of the Pareto front. However, an input can have a high information gain even if it is not Pareto dominating other inputs, but it provides high information about a new region in the search space. This conflicting issue does not occur with HVI, since HVI is always monotonically increasing with respect to Pareto dominance.

Nevertheless, we provide an ablation study comparing variants of our acquisition functions with EHVI, MESMO, and JESMO based scalarization

We present the results of our ablation study comparing the performance of the BINOM acquisition function using Expected Hypervolume Improvement (EHVI), Max-value Entropy Search (MES), and Joint Entropy Search (JES) with horizon lengths of 2, 4, and 6, across six real-world benchmark problems. For clarity, we refer to the BINOM variant using EHVI as simply BINOM. The hypervolume plots for each benchmark illustrate that BINOM-EHVI generally outperforms the other acquisition functions, demonstrating its superiority in guiding the optimization process towards a high-quality Pareto front. Even in instances where BINOM-EHVI does not achieve the best performance, it consistently surpasses our myopic baseline methods, reaffirming the robustness and effectiveness of the BINOM approach in diverse optimization scenarios.

### A.4 Synthetic Benchmarks

In addition to the benchmarks presented in the main paper, we evaluated our method on the ZDT3 benchmark in Figure 5. The ZDT3 problem is a widely used test in multi-objective optimization literature (Konakovic Lukovic et al., 2020).

To demonstrate the effectiveness of our non-myopic methods on problems with a larger number of objectives, we evaluated our most computationally efficient method, BINOM, on various DTLZ benchmark problems (Deb et al., 2005). Figure 6 presents results for DTLZ-1 (d=11, K=4), DTLZ-1 (d=7, K=5), DTLZ-2 (d=8, K=4), and DTLZ-3 (d=9, K=5).

### A.5 Additional Baselines

**HVKG** (Daulton et al., 2023). HVKG is notable for its cost-aware decoupled strategy, which assigns specific costs to each objective function and avoids evaluating all objectives in every iteration. To ensure a fair

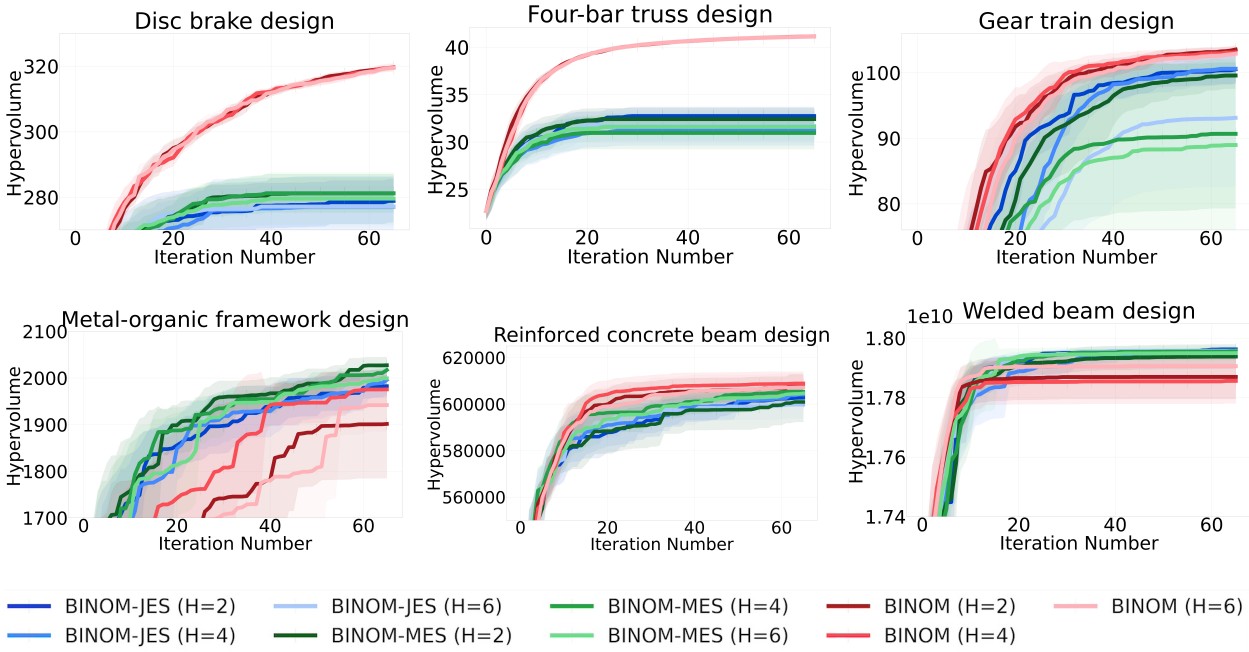

Figure 4: Hypervolume results for BINOM with MESMO (BINOM-MES) , JESMO (BINOM-JES), and EHVI (BINOM) acquisition functions with lookahead horizon $H \in \{2, 4, 6\}$.

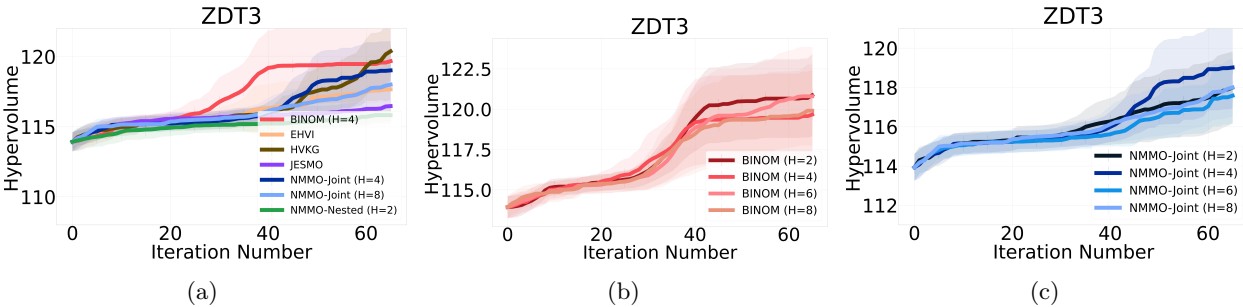

Figure 5: With respect to the ZDT3 problem, Figure 5a shows the comparison of all baselines, Figure 5b shows the hypervolume comparison for BINOM with varying horizon lengths, and Figure 5c depicts the hypervolume comparison for NMMO-joint with different horizon lengths.

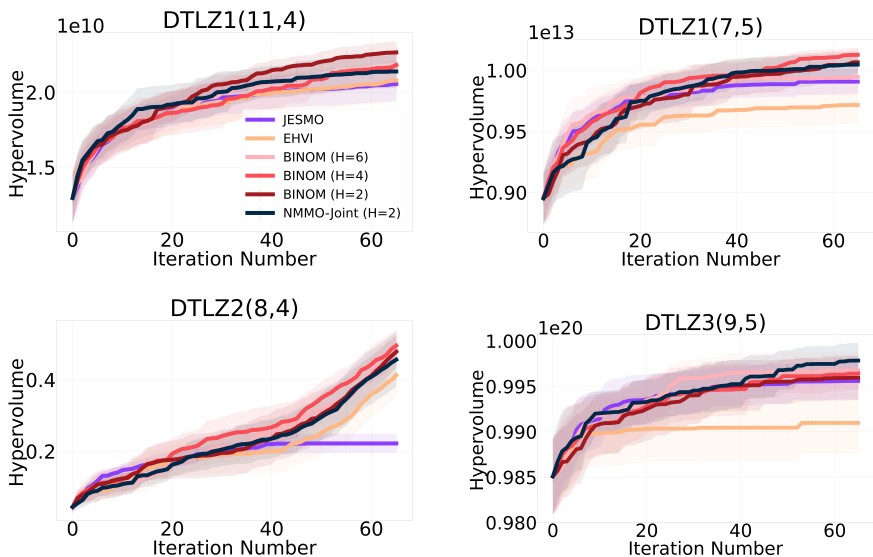

Figure 6: Hypervolume results for variants of the DTLZ problem.

comparison between HVKG and our NMMO approaches, we adjusted the objective function costs to zero for all problems in our experiments, which allows all methods to operate under similar cost conditions. Moreover, to accurately measure the hypervolume indicator, we implemented a procedure where all objective functions of the selected inputs under the HVKG method are evaluated at each iteration. These expensive evaluations are then used to calculate the hypervolume, ensuring that all methods' performances are assessed similarly.

**NMMO-Nested.** We include the NMMO-Nested with $H = 2$ results in Figure 7. Due to computational cost, we were not able to run experiments of NMMO-Nested with additional horizon values. As explained in section 5, while the NMMO-Nested method should theoretically outperform the other non-myopic baselines, its optimization is very time-consuming and inefficient in practice, leading to slightly subpar results.

**MESMO.** We included MESMO (Max-value Entropy Search for Multi-objective Optimization (Belakaria et al., 2019)) in our baseline comparisons to assess its performance on various benchmarks. MESMO employs an advanced output space entropy search strategy, aiming to maximize the information gained about the Pareto front with each iteration. Despite its innovative approach, MESMO has been somewhat overshadowed in recent evaluations by newer methods like JESMO, which combines the strengths of both output and input space entropy searches. We include a comparison of how MESMO's performance stacks up against other state-of-the-art multi-objective Bayesian optimization techniques.

We use implementations from the BoTorch python package [*] for all baselines (JESMO, EHVI, MESMO, HVKG).

### A.6   Run times

The following are the computational complexities of our proposed methods.

*BINOM*: $O(KN^2H)$ Where K is the number of objectives, N is the number of observations, and H is the lookahead horizon.

*NMMO-Joint*: $O(KN^2H^2)$ This method has an additional factor of H due to the joint optimization over the lookahead horizon.

*NMMO-Nested*: $O(KN^2H^2M)$ Where M is the size of the discretized input space used for the nested optimization.

---

[*]https://github.com/pytorch/botorch

Table 1: Benchmark details.

| Problem name | d | K | r |
|---|---|---|---|
| Disc Brake Design | 4 | 3 | (6.1356, 6.3421, 12.9737) |
| Four Bar Truss Design | 4 | 2 | (2967.0243, 0.0383) |
| Gear Train Design | 4 | 3 | (6.6764, 59.0, 0.4633) |
| Reinforced Concrete Beam Design | 3 | 2 | (703.6860, 899.2291) |
| Welded Beam Design | 4 | 3 | (202.8569, 42.0653, 2111643.6209) |
| Metal-Organic Framework Design | 7 | 2 | (1.0, 1.0) |
| ZDT-3 | 9 | 2 | (11.0, 11.0) |

For context, the baseline EHVI method typically has a complexity of $O(KN^2)$.

While our non-myopic methods do introduce additional computational overhead compared to myopic approaches, the increase is not prohibitively high, especially considering the performance gains achieved and its intended use in budget-limited settings for scientific applications.

Table 2, shows the average run-time per iteration of each baseline on multiple benchmarks. We utilized an NVIDIA Quadro RTX 6000 GPU with 24,576 MiB memory capacity. We provide the average and standard deviation of 65 iterations in one run of the algorithm (in seconds). It is noteworthy that while the non-myopic methods take longer, they also provide better performance with a limited budget.

Table 2: Runtime comparison (seconds per iteration) rounded to the closest integer.

| Method | H | Reinforced concrete beam design | Four-bar truss design | Gear train design | Metal-organic framework design |
|---|---|---|---|---|---|
| EHVI | - | $1 \pm 0$ | $0 \pm 0$ | $1 \pm 0$ | $5 \pm 0$ |
| JESMO | - | $2 \pm 2$ | $4 \pm 3$ | $3 \pm 3$ | $6 \pm 0$ |
| HVKG | 1 | $47 \pm 20$ | $32 \pm 5$ | $52 \pm 14$ | $102 \pm 27$ |
| BINOM | 2 | $4 \pm 2$ | $9 \pm 3$ | $13 \pm 3$ | $17 \pm 2$ |
|  | 4 | $7 \pm 4$ | $22 \pm 9$ | $24 \pm 3$ | $17 \pm 1$ |
| NMMO-Joint | 2 | $28 \pm 13$ | $22 \pm 5$ | $59 \pm 10$ | $47 \pm 7$ |
|  | 4 | $52 \pm 19$ | $35 \pm 15$ | $75 \pm 10$ | $55 \pm 4$ |
|  | 8 | $104 \pm 33$ | $125 \pm 41$ | $181 \pm 52$ | $73 \pm 13$ |
| NMMO-Nested | 2 | $560 \pm 58$ | $802 \pm 62$ | $616 \pm 151$ | $754 \pm 141$ |

## A.7 Additional Experimental Details

*Joint Entropy Search for Multi-Objective Optimization (JESMO)*: We use the JESMO (Tu et al., 2022) algorithm as implemented in the BoTorch python package (Balandat et al., 2020) as a baseline for comparison in our study. However, we encountered a significant challenge with this approach: the JESMO implementation in BoTorch failed almost immediately when applied to many of our real-world multi-objective problems. This failure was primarily due to the algorithm's inability to identify a sufficient number of optimal Pareto sample points. Specifically, it struggled to find the required 10 sample points. To accommodate this limitation and ensure the feasibility of using JESMO in our experiments, we adjusted the requirement to a smaller set of 4 optimal Pareto sample points. We determined the optimal number of Pareto sample points by monitoring

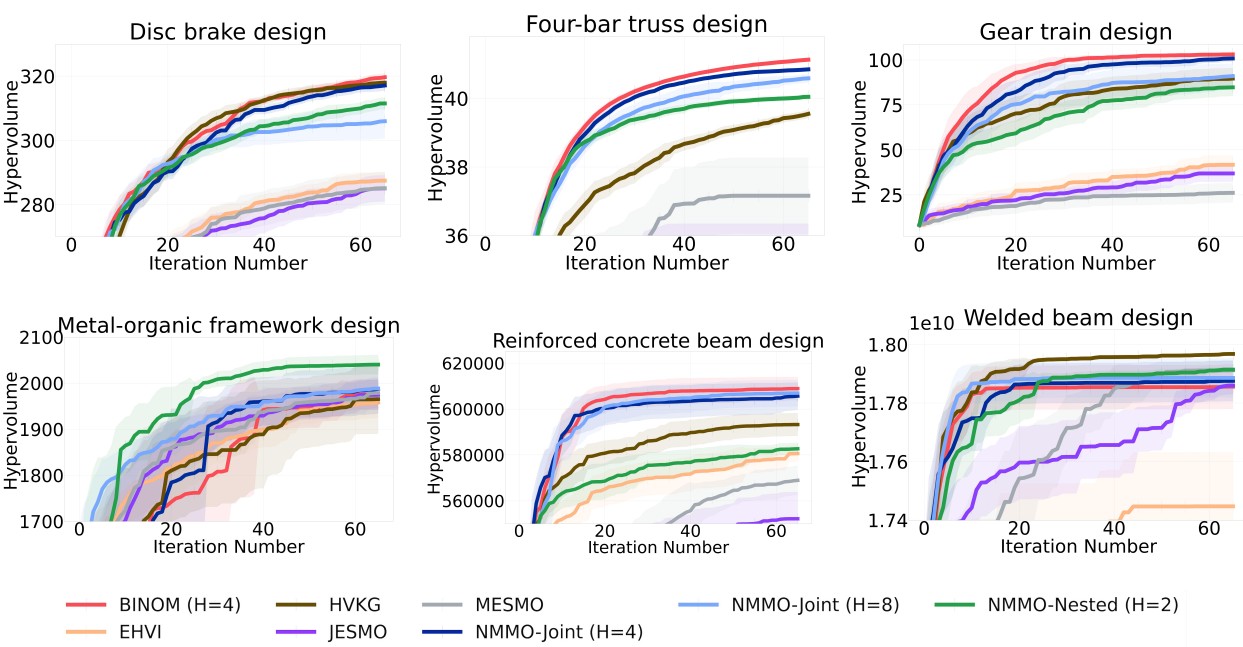

Figure 7: Hypervolume results on real-world benchmarks including additional baselines

the instances where JESMO failed to identify a sufficient number of optimal points across various runs. This empirical approach allowed us to adjust the number of required sample points to a level where the algorithm could consistently perform without failure.

### A.8 Limitations and Future Work.

Despite the proposed advancements in solving the finite-horizon multi-objective SED problem using hypervolume improvement strategies, several limitations remain in our approach, opening the way for future improvements. While the introduction of alternative lower bounds and the use of BEHVI are designed to provide a tractable alternative to solving the Bellman equation, these strategies raise questions about the tightness of the lower bounds and its ability to accurately represent the intractable optimal policy. This question is closely related to prior work on the adaptivity gap (Krause & Guestrin, 2007). Given that we established the possibility of using the Bellman equation with HVI, a promising future direction would be to build a multi-step expected HVI approach analogous to the multi-step expected improvement (Jiang et al., 2020b).

## B Hypervolume Improvement Additivity Proof

Let $\mathcal{Y}_0$ be the initial Pareto front, and let $(\mathbf{x}_t, \mathbf{y}_t)$ be the new data point added at time step $t$, where $t \in [1, T]$. We denote the updated Pareto front at time step $t$ as $\mathcal{Y}_t$. The hypervolume improvement at time step $t$ is given by:

$$HVI_t = HVI(\mathbf{y_t}|\mathcal{Y}_{t-1}) = HV(\mathcal{Y}_{t-1} \cup \mathbf{y}_t) - HV(\mathcal{Y}_{t-1}) = HV(\mathcal{Y}_t) - HV(\mathcal{Y}_{t-1}) \tag{17}$$

Now, let's consider the final hypervolume improvement after $T$ time steps, we denote the final hypervolume improvement as $HVI_{Total}$

$$HVI_{Total} = HV(\mathcal{Y}_T) - HV(\mathcal{Y}_0)$$

We want to prove that:

$$HVI_{Total} = \sum_{t=1}^{t=T} HVI_t$$

By expanding the summation, we get a telescoping series:

$$\sum_{t=1}^{t=T} HVI_t = \sum_{t=1}^{t=T} HV(\mathcal{Y}_t) - HV(\mathcal{Y}_{t-1}) \tag{18}$$

$$= (HV(\mathcal{Y}_1) - HV(\mathcal{Y}_0)) + (HV(\mathcal{Y}_2) - HV(\mathcal{Y}_1)) + \cdots + (HV(\mathcal{Y}_T) - HV(\mathcal{Y}_{T-1})) \tag{19}$$

The intermediate terms of the series canceling out lead to:

$$\sum_{t=1}^{t=T} HVI_t = HV(\mathcal{Y}_T) - HV(\mathcal{Y}_0) \tag{20}$$

Therefore,

$$HVI_{Total} = \sum_{t=1}^{t=T} HVI_t \tag{21}$$

This proves that the final hypervolume improvement is indeed the summation of hypervolume improvements occurring at individual time steps.

The intuition behind this proof is that the hypervolume improvement at each time step represents the additional volume dominated by the updated Pareto front compared to the previous Pareto front. By summing up these individual improvements, we account for the total volume gained from the initial Pareto front to the final Pareto front.

