# OpenReview forum: "Non-Myopic Multi-Objective Bayesian Optimization"
_TMLR — Accepted by TMLR_

### Review · Reviewer_GEYU · 2025-01-18

**Summary Of Contributions:**

The paper proposes non-myopic algorithms for multi-objective Bayesian optimization. It uses expected hypervolume improvement as the scalarization metric with previous non-myopic algorithms for single-objective Bayesian optimization. It proposes 3 algorithms, NMO-Nested, NMO-Joint, and BINOM, listed in increasing levels of approximation error (and decreasing levels of computational complexity). The algorithms are compared to several baselines with several objective functions.

**Audience:**

Yes

**Broader Impact Concerns:**

None.

**Claims And Evidence:**

Yes

**Requested Changes:**

**Would strengthen work**:
1. It is not clear what Equation (1) means. How is this the Pareto hypervolume indicator with respect to a reference $r$ as described in the previous paragraph?
2. In the first paragraph of page 4, should Equation 5 refer to Equation 6?
3. The citation Osbourne & Osbourne (2010) is wrong, both refer to the same person.
4. In lines 6 and 8 of Algorithm 1, the argmax's should include all arguments to the function (other than $D_t$), otherwise the statement is not well-defined.

**Strengths And Weaknesses:**

**Strengths**:
1. The experimental results suggest that the proposed non-myopic algorithms significantly outperform baselines in the tested settings.
2. The experiments are comprehensive, with many objective functions, runtime analyses, and ablation studies.
2. The paper is written clearly. The ideas developed are straightforward to follow.

**Weaknesses**:
1. The paper lacks significant novelty. The algorithms are largely straightforward combinations of ideas developed in previous non-myopic and multi-objective BO papers.

Overall, the paper is likely to be of interest to BO practitioners dealing with multi-objective BO problems, while it is of little interest from a theoretical perspective.

---

> ### Author Response · Authors · 2025-02-12
>
> Thank you for your review and constructive feedback. Below we try to address your concerns.
>
> **W1** - We agree that our work builds on the single-objective findings related to the lower bound on the Bellman equation. However, applying these principles to the multi-objective setting presents important challenges that we explicitly address in the paper. Specifically, multi-objective optimization requires handling scalarization and Pareto optimality in a way that differs fundamentally from the single-objective case. Extending the lower-bound formulation and incorporating hypervolume improvement (HVI) within a non-myopic framework for MOO is not straightforward.
> To highlight the novelty of our approach, we also discuss how several other scalarization methods that might appear additive still fail to satisfy the monotonicity condition with respect to Pareto optimality. Furthermore, our method is the first to propose a non-myopic policy for MOO using HVI, establishing a foundation for future advancements in multi-objective Bayesian optimization. We hope that this contribution not only enhances current methodologies but also opens new research directions in this field.
>
> **RC1** - We appreciate the reviewer's suggestion. We adjusted the notation of the equation to indicate that the max is over a vector. This equation defines the multi-step optimization problem over a vector of functions. The intended meaning of Equation (1) is explained in the Non-myopic MOO problem paragraph. We would like to clarify that this equation is a general multi-objective optimization definition and does not include a definition of the hypervolume. We added a formal definition of the hypervolume to avoid the confusion.
>
> **RC2** - Yes, thank you for pointing this typo. We updated the manuscript to reflect that.
>
> **RC3** - We corrected the citation.
>
> **RC4** - We adjusted these lines to be more precise.

---

### Review · Reviewer_uVHU · 2025-01-29

**Summary Of Contributions:**

This paper proposes a novel acquisition function for multi-objective black-box optimization that is non-myopic—arguably the first in this setting. Specifically, the authors explore the use of Hypervolume Improvement as a utility function for optimal sequential decision-making. Building on the framework of [Shali Jiang et al. (ICML 2020)](https://proceedings.mlr.press/v119/jiang20b/jiang20b.pdf), they derive a *lower bound* on the true expected utility. The paper discusses three strategies for solving the multi-step Bellman equation:
- Nested Non-Myopic Multi-Objective Acquisition Function – provides an exact computation of the lower bound.
- Joint Non-Myopic Multi-Objective Optimization – a scalable alternative to the previous approach.
- Batch-Informed Non-Myopic Multi-Objective Optimization – a fast, approximate version of Expected Hypervolume Improvement (EHVI).

The proposed methods are evaluated on at least six real-world experimental design problems, with promising results. Additionally, empirical results for different horizons and a computational runtime comparison are presented. An ablation study of information-gain within the proposed framework is also investigated.

**Audience:**

Yes

**Claims And Evidence:**

Yes

**Requested Changes:**

- Avoid making claims about information gain in this setting. There is no compelling evidence demonstrating that information gain is monotonic or additive in this context, and including such suggestions weakens the main point of this paper. I would adhere to the principle of "one idea, one paper." If you intend to propose a new information-theoretic-inspired acquisition function, it should be thoroughly evaluated and discussed in depth, and not in two paragraphs.

- Consider improving the mathematical definitions. For example, the function $\alpha(x)$ is overloaded so many times that it gets confusing. Consider adding a new terminology for each definition such as $\alpha_{BINOM} (x)$. You can use a similar strategy that the authors (Shali Jiang et al.) used in equation 5 of this paper [Efficient Nonmyopic Active Search (ICML 2017)](https://proceedings.mlr.press/v70/jiang17d/jiang17d.pdf). In fact, this paper I think was the first to point out that you can do a batch selection to approximate a non-myopic sequential decision making (it assumes independency when you do so).

- Correct the citations by using \citep and \citet appropriately and consistently.

- In Figure 1, either use a log scale for the bottom row or normalize all objectives for consistency.

- Address typographical errors, particularly in Appendix B, where they are prevalent.

- In Section 5, the evaluation metric should be referred to as Hypervolume, not HVI.

- I don't see a formal definition of hypervolume.

- Equation 1 is not precise. $f_1(x) ... f_2(x) $ could be anything. Specifically, a casual reader my infer that this is an addition.

- $D_t$ is never defined as a set of $t$ observations $(x, y)$. In some contexts, *observation* is synonymous of the response variable $y$.

**Strengths And Weaknesses:**

*Strengths*

- This research project is highly relevant to the community. Multi-objective black-box optimization is an important topic for both researchers and practitioners, and the introduction of a non-myopic acquisition function in this setting is a valuable contribution.

- The derivation of the proposed acquisition function is well contextualized within previous work and is presented in a relatively clear and easy-to-follow.

- The selection of benchmark problems is particularly interesting. They look like compelling design problems that could significantly benefit from multi-objective optimization.

*Weaknesses*

The primary weakness of this paper is the clarity of its presentation. The mathematical expressions require greater precision, and all elements critical to the proposed method should be explicitly defined. I strongly recommend a thorough and detailed revision of the presentation to enhance its clarity.

---

> ### Author Response · Authors · 2025-02-12
>
> Thank you for your review and constructive feedback. Below we try to address your concerns.
>
> **W1** - To enhance the presentation and precision of our work, we conducted a thorough revision of the paper in which we incorporated your suggestions. We submitted a new revised version of the paper.
>
> **RC1** - We appreciate the reviewer’s suggestion and would like to clarify that we are not claiming information gain as a contribution of this work. Rather, we intended to include it as an ablation study to demonstrate that information gain-based scalarizations do not satisfy the necessary conditions in our formulation. Specifically, we discuss that they do not maintain monotonicity with respect to Pareto front quality. To avoid any confusion and to ensure a more focused presentation on our contributions, we moved this discussion to the ablation study section in the Appendix, where it will serve as a supplementary discussion rather than a main point of the paper. This adjustment maintains a clear emphasis on our core contributions.
>
> **RC2** - We appreciate the reviewer’s suggestion. We made the suggested changes to the notation. Specifically, we introduced distinct notation, such as $\alpha_{BINOM}$, $\alpha_{Joint}$, and $\alpha_{Nested}$ to avoid overloading functions and enhance clarity. We also took inspiration from the notation strategy used in the mentioned paper to improve readability. Additionally, we acknowledged this work in both the problem setup section and the related work.
>
> **RC3** -  We updated the citations to address this issue.
>
> **RC4** -  Thank you for highlighting this. The y-axis in Figure 1 was intentionally scaled to showcase the distinctions among the non-myopic methods, which would appear visually overlapped on the full scale due to their substantial performance improvement over the baselines.
>
> **RC5,6** - We addressed the typos in the Appendix and made the suggested change to Section 5.
>
> **RC7** - In the initial version of the paper, we provide a brief explanation of hypervolume in the problem setup section, along with the original citation. To ensure completeness and clarity, we added a formal definition of hypervolume in the main text.
>
> **RC8** - We adjusted the notation of this equation to indicate that the max is over a vector and avoid confusion. Additionally the intended meaning of this equation is explained in the Non-myopic MOO problem paragraph right before the equation.
>
> **RC9** -  We added a definition of $D_t$.

---

### Review · Reviewer_eCCK · 2025-02-06

**Summary Of Contributions:**

This paper introduces a novel method for non-myopic multi-objective Bayesian optimization (BO). The paper firstly identifies the requirements on the reward/utility function in order to satisfy the Bellman optimality principle in non-myopic BO, and then proposes to use HVI as the scalarization approach which satisfies these requirements. Several methods are proposed which differ in terms of the degree of approximation and the trade-off between computation and performance. Experimental results show that the proposed methods perform better than classical myopic BO methods.

**Audience:**

Yes

**Broader Impact Concerns:**

I do not have any concern about the ethical implications of the work.

**Claims And Evidence:**

Yes

**Requested Changes:**

A few places may require some clarifications:
- Equation (1), it is unclear what is the objective being maximized here.
- Section 4.2.1 and Equation (10), what is $\mathcal{Y}$ here? How is it chosen?
- In Section 4.2.2., unless I missed it, I think the three variants of the proposed method are not clearly discussed in this section, more specifically, which algorithm name corresponds to which version. It is only discussed in Algorithm 1.

**Strengths And Weaknesses:**

Strengths:
- The paper is nicely motivated and clearly discusses the challenges faced by non-myopic multi-objective BO.
- The paper is well written, firstly listing the requirements on the scalarization approach and then proposing a scalarization method which satisfy these requirements.
- It is commendable that the paper proposed several variants of their main algorithm, which offer different trade-offs between performance and computation. This offers practitioners significant flexibility when adopting their proposed method.
- The paper provides discussions at many places, which are very useful for helping the reader understand the essential insights about the method and experimental findings.

Weaknesses:
- In the experiments, the maximum number of objectives $K$ is 3. I think it would help reinforce the empirical conclusions from the paper if the proposed methods are applied to problems with a larger number of objectives.
- Just to clarify, does BINOM incur less computation or more computation than NMMO-Joint? Why is NMMO-Joint tested using a larger horizon than BINOM?

---

> ### Author Response · Authors · 2025-02-12
>
> Thank you for your review and constructive feedback.  Below we try to address your concerns.
>
> **W1** - We appreciate the reviewer’s suggestion. It is important to note that we used a wide variety of standard benchmarks for MOO in line with most papers on multi-objective Bayesian optimization (MOBO) that typically experiment with 2–3 objectives. That said, we agree that demonstrating the performance of our method on problems with a higher number of objectives would help reinforce our empirical conclusions.  To address this, we are currently running experiments with a larger number of objectives. We will add these experiments as soon as they are done. However, due to the computational cost, these experiments might require more than two weeks to complete.
>
> **W2** - That is correct. BINOM incurs less computational cost than NMMO-Joint. However, since NMMO-Joint is a tighter approximation of the lower bound, we wanted a more comprehensive performance comparison for NMMO-Joint.
>
> **RC1** - We adjusted the notation of this equation to indicate that the max is over a vector and avoid confusion. Additionally the intended meaning of this equation is explained in the Non-myopic MOO problem paragraph right before the equation.
>
> **RC2** - $\mathcal{Y}$ represents a set of evaluated objective function value vectors that define a Pareto front. For example, at an iteration $t$ of the algorithm, it consists of all non-dominated output vectors out of the previously observed function evaluations obtained from queried inputs. Equation 10 defines Hypervolume Improvement (HVI) as a measure of the contribution of a newly evaluated solution added to the Pareto front in multi-objective optimization.
>
> **RC3** - We appreciate the reviewer’s suggestion. We made the suggested changes to the notation. Specifically, we introduced distinct notation, such as $\alpha_{BINOM}$, $\alpha_{Joint}$, and $\alpha_{Nested}$ to enhance clarity.

---

> > ### Author Response · Authors · 2025-02-19
> >
> > We have added experimental results on synthetic benchmarks with a larger number of objectives in the appendix (Figure 6). Due to the higher computational cost of the other methods, these experiments were primarily conducted using BINOM, which is the most efficient to run. Additionally, we are currently running experiments with NMMO-Joint on these benchmarks and will include the results in the final version of the paper. However, due to their longer runtime, they did not complete before the rebuttal deadline.

---

### Decision · Action_Editor_WzMx · 2025-04-02

**Recommendation:** Accept as is

**Comment:**

A summary of the strength and weaknesses based on the reviews and rebuttal is provided below:

**STRENGTHS**

(1) The problem of nonmyopic MOBO and its challenges are well motivated.

(2) The paper is well-written and incorporates important insights and experimental findings.

(3) Several variants of the main algorithm are proposed, each with a different trade-off between performance vs. computation, allowing users to have the flexibility of options.

(4) The experimental results show that the proposed algorithms outperform the baselines.


**WEAKNESSES**

(1) Experimental results on a larger number of objectives are initially lacking. The authors have provided them in the rebuttal to address this concern.

(2) There were initial concerns on clarity for which the authors have clarified in the rebuttal.

Since the main concerns of the reviewers have been addressed, the pros outweigh the cons.


For completeness, the authors can consider citing and commenting on the following earlier references on nonmyopic BO:

Nonmyopic Gaussian Process Optimization with Macro-Actions. AISTATS 2020.

Sequential Bayesian optimisation for spatial-temporal monitoring. UAI 2014.

Gaussian process planning with Lipschitz continuous reward functions: Towards unifying Bayesian optimization, active learning, and beyond. AAAI 2016.

**Audience:**

Yes, the Bayesian/blackbox optimization and derivative-free optimization communities would be interested in the findings of this work.

**Claims And Evidence:**

Yes, after the rebuttal, the claims made in the submission are well-supported by evidences.